# Tracking progress towards malaria elimination in China: Individual-level estimates of transmission and its spatiotemporal variation using a diffusion network approach

Isobel Routledge[1]☉*, Shengjie Lai[2]☉, Katherine E. Battle[3], Azra C. Ghani[1], Manuel Gomez-Rodriguez[4], Kyle B. Gustafson[5], Swapnil Mishra[1], Juliette Unwin[1], Joshua L. Proctor[5], Andrew J. Tatem[2], Zhongjie Li[6], Samir Bhatt[1]

1 Imperial College London, London, United Kingom, 2 University of Southampton, Southampton, United Kingdom, 3 University of Oxford, Oxford, United Kingdom, 4 Max Planck Institute for Software Systems, Saarbrücken, Germany, 5 Institute for Disease Modelling, Bellevue, Washington, United States of America, 6 Chinese Centers for Disease Control and Prevention, Beijing, China

☉ These authors contributed equally to this work.
* i.routledge15@imperial.ac.uk

**Data Availability Statement:** The estimated maximum-a-posteriori Rc estimates and results of

## Abstract

In order to monitor progress towards malaria elimination, it is crucial to be able to measure changes in spatio-temporal transmission. However, common metrics of malaria transmission such as parasite prevalence are under powered in elimination contexts. China has achieved major reductions in malaria incidence and is on track to eliminate, having reporting zero locally-acquired malaria cases in 2017 and 2018. Understanding the spatio-temporal pattern underlying this decline, especially the relationship between locally-acquired and imported cases, can inform efforts to maintain elimination and prevent re-emergence. This is particularly pertinent in Yunnan province, where the potential for local transmission is highest. Using a geo-located individual-level dataset of cases recorded in Yunnan province between 2011 and 2016, we introduce a novel Bayesian framework to model a latent diffusion process and estimate the joint likelihood of transmission between cases and the number of cases with unobserved sources of infection. This is used to estimate the case reproduction number, $Rc$. We use these estimates within spatio-temporal geostatistical models to map how transmission varied over time and space, estimate the timeline to elimination and the risk of resurgence. We estimate the mean $Rc$ between 2011 and 2016 to be 0.171 (95% CI = 0.165, 0.178) for *P. vivax* cases and 0.089 (95% CI = 0.076, 0.103) for *P. falciparum* cases. From 2014 onwards, no cases were estimated to have a $Rc$ value above one. An unobserved source of infection was estimated to be moderately likely (p>0.5) for 19/ 611 cases and high (p>0.8) for 2 cases, suggesting very high levels of case ascertainment. Our estimates suggest that, maintaining current intervention efforts, Yunnan is unlikely to experience sustained local transmission up to 2020. However, even with a mean

the additive regression analysis are included with the Supporting Information, as well as all code required to calculate Rc and generate the manuscript figures. The underlying patient line-list data used in the study are available from The Chinese Center For Disease Control and Prevention, 155 Changbai Road Changping District, Beijing 102206,China. Tel: +86-10-58900240, 58900216 or from Dr Junling Sun (email: sunjl@chinacdc.cn ). She is the Head of the Branch of Parasitic Disease, Division of Infectious Disease, China CDC.

**Funding:** This work was funded by a studentship to IR from the Wellcome Trust (109310/Z/15/Z). IR, SB, AG and JU additionally acknowledge Centre support (MR/R015600/1; MRC Centre for Global Infectious Disease Analysis, School of Public Health, Imperial College London) jointly funded by the UK Medical Research Council (MRC) and the UK Department for International Development (DFID) under the MRC/DFID Concordat agreement and is also part of the EDCTP2 programme supported by the European Union. The funders had no role in study design, data collection and analysis, decision to publish, or preparation of the manuscript.

**Competing interests:** The authors have declared that no competing interests exist.

of 0.005 projected up to 2020, locally-acquired cases are possible due to high levels of importation.

## Author summary

Although malaria is still responsible for a great deal of death and illness in many parts of the world, many national control programmes have made great strides in controlling malaria and now are in a position to aim for elimination. However, in order to monitor progress towards elimination and plan interventions, it is crucial to measure malaria transmission and how it varies over space and time. However, traditional metrics used to measure malaria transmission are not suitable in elimination settings. China is one example of a country approaching elimination, with aims to eliminate the disease by 2020. Using a detailed individual level dataset of the times and locations of people showing symptoms of malaria, we use approaches adapted from the study of how information spreads through social networks to estimate the likelihood of transmission occurring between cases. This information is used to estimate how many people we expect each case to go on to infect. In elimination settings, this number is an indication of how quickly elimination will be reached and how likely we are to see a resurgence in cases once elimination is achieved. Our results show a decline in this metric over time, as well as seasonal changes in transmission which are different to the patterns in when the most cases were observed.

## Introduction

In 2016 the World Health Organisation listed 21 countries for whom it would be realistic to achieve elimination of malaria by 2020, defined as zero indigenous cases over three consecutive years [1]. The largest of these is the People's Republic of China (thereafter called China). In 2017 China reported no indigenous malaria cases for the first time since malaria became a notifiable disease in 1956 [2,3]. The country has experienced a major decline in the burden of malaria, from an annual incidence of 24 million cases (2961 cases per 100,000) in 1970 [4]. This reduction has been attributed to a combination of socioeconomic improvements and the scale-up of interventions to control malaria [5]. In 2010, China set out an ambitious plan for the national elimination of malaria by 2020 (the National Malaria Elimination Programme, NMEP). Elements of the plan included improved surveillance, timely response, more effective and sensitive risk assessment tools and improved diagnostics [6]. A key policy change implemented in 2010 as part of the NMEP was the introduction of the 1-3-7 system: aiming for case reporting in one day, which is then investigated within three days, with a focused investigation and action taken in under seven days [7].

Although China is making rapid progress towards this goal, 2,675 imported cases were reported in 2017, highlighting the risk of re-introduction [3]. Large numbers of people move between China and malaria endemic countries, both from sub-Saharan Africa and from South East Asia [8,9], driven by tourism and Chinese oversea investment [10]. Concerns remain about re-emergence of malaria, which has occurred several times in the early 2000s as a result of importation and favourable climatic conditions for competent vectors [11]. Therefore, in order to achieve three consecutive years of zero indigenous cases (the requirement for WHO

certification of elimination), a sustained and targeted investment in surveillance together with efficient treatment is necessary.

Yunnan province has recorded malaria outbreaks and remains an identified foci of residual transmission as other areas in the country have reached elimination [12–16]. The province shares borders with Myanmar, Vietnam and Laos and has a strong agricultural focus. Previous studies suggest that seasonal agricultural workers and farmers are at highest risk of contracting malaria in Yunnan, with rice yield and the proportion of rural employees being spatial factors positively associated with malaria incidence [17]. The border region of Myanmar and Yunnan is generally ecologically suitable for malaria transmission, has a large mobile population, with few natural geographic borders separating the two countries, as well as being a site of socio-political conflict and instability [18]. In this context, it can be unclear if there is any sustained local transmission or if all the observed cases are the result of short, stuttering transmission chains following importation into suitable areas. As the area of highest concern for re-emergence in China and the last to reach zero cases, we therefore sought to characterise the transmission dynamics of both *Plasmodium vivax* and *Plasmodium falciparum* in the region as China approaches elimination certification.

Methods from outbreak analysis and network research have recently been developed and applied to quantify the transmission of malaria and other infectious diseases in near-elimination and epidemic settings [19–21]. In near elimination contexts with strong surveillance systems, traditional metrics of malaria such as parasite prevalence are not appropriate due to small numbers and extremely sparse and spatiotemporally heterogeneous distributions of infections. However due to the strength of the surveillance system in China, detailed information is available about each individual case (including the time of symptom onset and location of residence), and case reporting is believed to be very high. By adapting and applying a diffusion network approach [22] within a Bayesian framework, we quantify case reproduction numbers [23], $R_c$, and uncertainty in these estimates for all *P. vivax* and *P. falciparum* cases of malaria recorded in Yunnan province between 2011 and 2016. We incorporate these estimates into geostatistical models and an additive regression model to estimate how $R_c$ varied over space and time which we use to estimate timelines to elimination and likelihood of resurgence.

## Results

### $R_c$ estimates over time

Between 2011 and 2016, 3496 cases of probable and confirmed *P. vivax* infection including mixed infections were observed in Yunnan province (2881 imported, 615 locally acquired). Including mixed infections, 818 *P. falciparum* infections were observed, of which 75 were locally acquired. The mean $R_c$ value estimated for *P. vivax* during this period was 0.171 (95% CI = 0.165, 0.178) and 0.089 (95% CI = 0.076, 0.103) for *P. falciparum* cases (S1 Fig). We estimate a decline in $R_c$ over time for both *P. vivax* (Fig 1A and 1B) and *P. falciparum* (Fig 1C and 1D), with the most rapid declines occurring between 2012 and 2014 (Fig 1A and 1C). No $R_c$ values above one were observed after 2014 for either species. These findings are consistent with varying levels of uncertainty about the serial interval distribution (S2 Fig). Observationally, the seasonal patterns of $R_c$ estimates also differ from observed incidence (S3 Fig). The case count shows a clear seasonal peak in May which is not reflected in the estimated $R_c$ values.

### Unobserved sources of infection

For all cases, there is a competing hazard, $\varepsilon$, of infection from an unobserved source, for example an unreported case. For *P. vivax*, 19 out of 615 locally acquired cases were estimated to have a moderate chance of having an unobserved source of infection (defined as an estimated

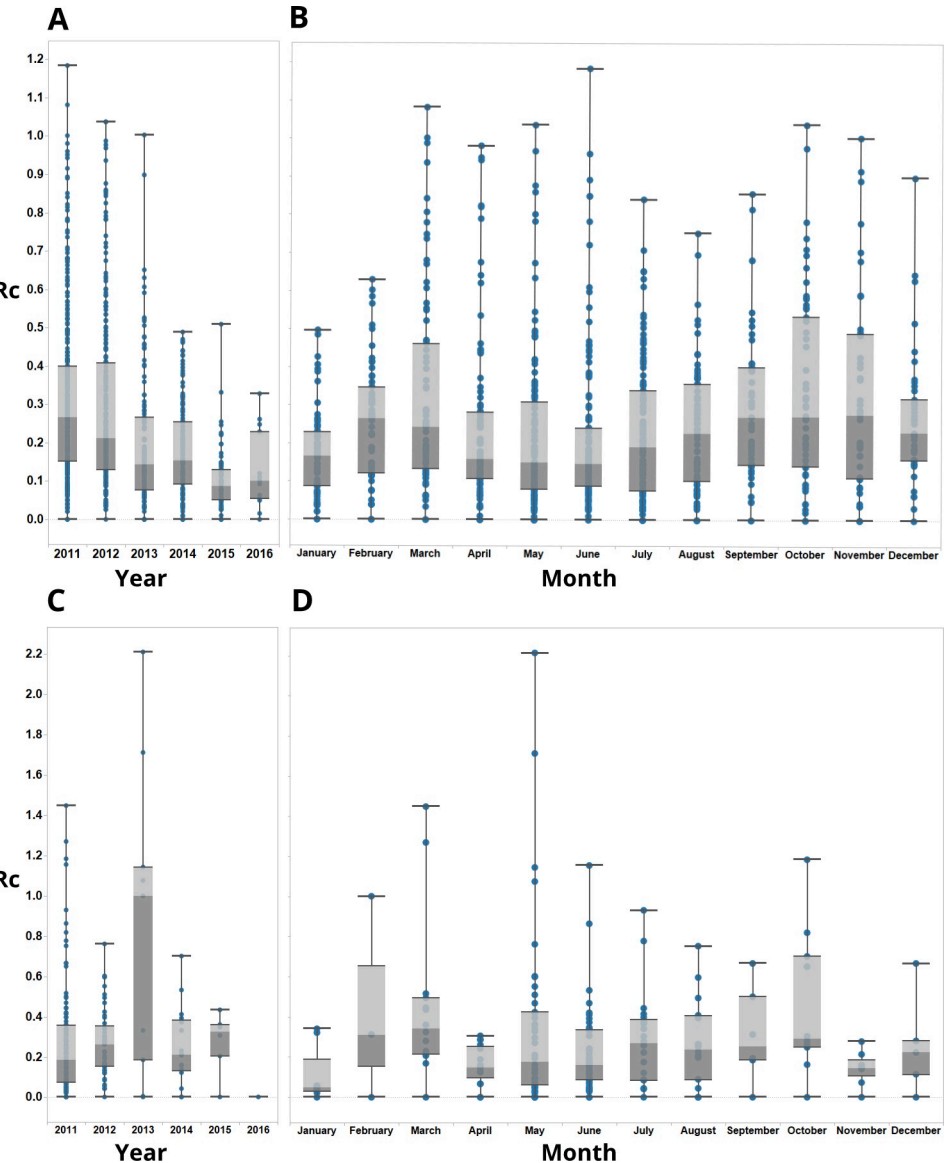

**Fig 1.** Boxplots showing $R_c$ estimates for *P. vivax* (A and B) and *P. falciparum* (C and D), aggregated by year (A and C) and month (B and D) of symptom onset. Points represent individual $R_c$ estimates. Boxplots show median, upper and lower quartiles for $R_c$ each.

epsilon edge, $\varepsilon$ of $0.5 \leq \varepsilon \leq 0.8$) and 2 cases were estimated to have a high chance of an unobserved source of infection (defined as an estimated epsilon edge, $\varepsilon$ of $\varepsilon \geq 0.8$). Together, this represents 3% of locally acquired cases with a moderate to high chance of external infection sources. For *P. falciparum*, 2 out of 75 local cases were estimated to have a high chance of having an unobserved source of infection (estimated $\varepsilon \geq 0.8$) and no other cases were estimated to have a moderate change of having an unobserved source of infection (S4 Fig).

## Spatial patterns of $R_c$

As transmission declined between 2011 and 2016, we observed a reduction in the incidence of locally-acquired cases which is reflected in a reduction in our estimates of the reproduction

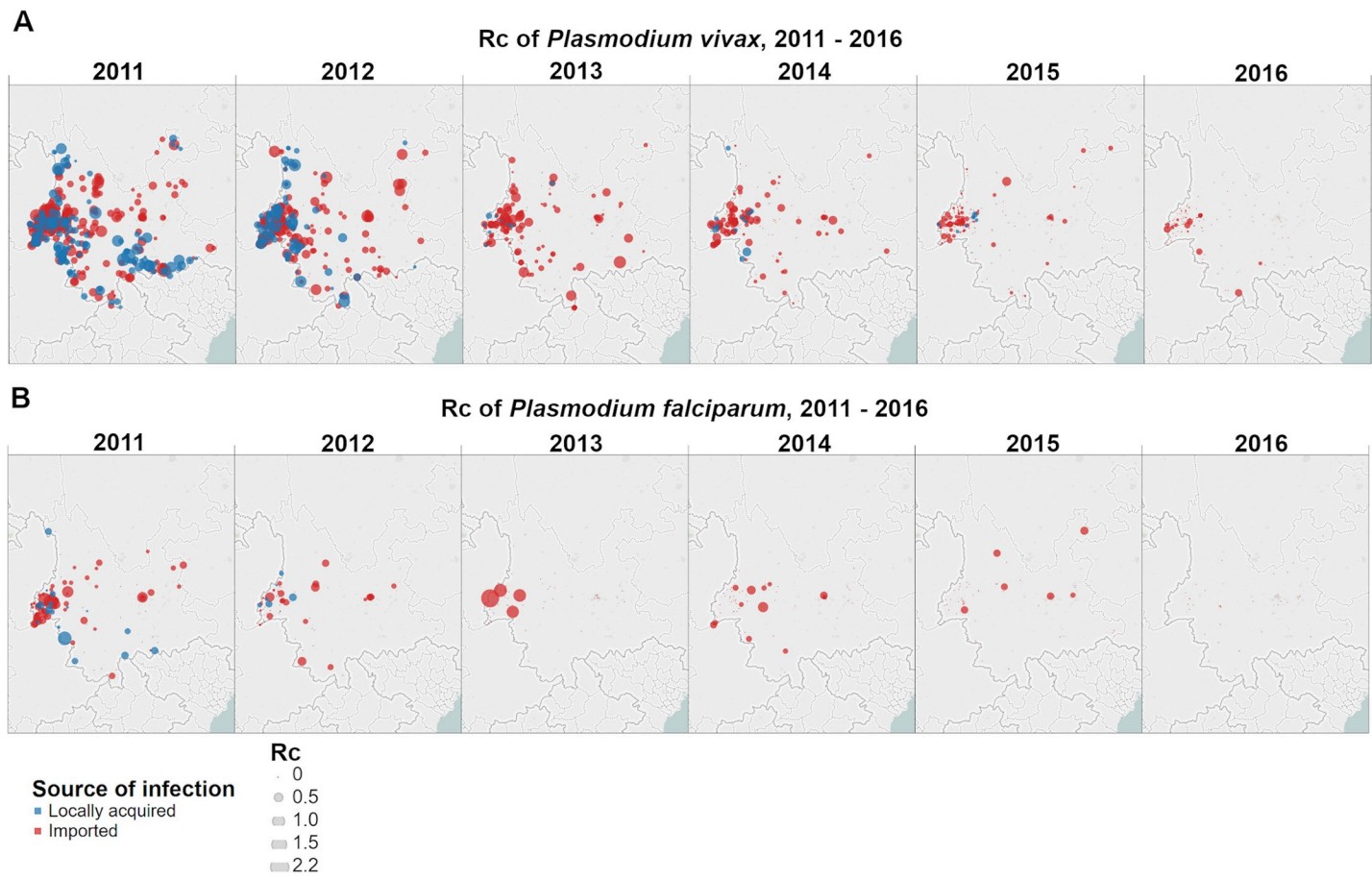

**Fig 2.** Map of $R_c$ estimates by year for A) *P. vivax* and B) *P. falciparum*. Blue points represent locally acquired cases; red points represent imported cases. The diameter of the point represents the size of the $R_c$ estimate. Base map of administrative boundaries come from Open Street Map and its contributors.

number of each locally-acquired case for both species (Fig 2). We estimate a decline in the probability of a reproduction number for a *P. vivax* case being above zero over this period (Fig 3A and 3B), with the central parts of the province being the first to reach lower risks of non-zero $R_c$. The border area neighbouring Myanmar, where most cases were observed, had the lowest amount of uncertainty in the estimates. *P. falciparum* shows a decline in risk of $R_c > 0$ across the province, with the more isolated areas in the north of the province showing both the highest predicted risk but also the most uncertainty, due to a lack of cases observed there (S5 Fig). The main sources of uncertainty were small data sizes (Bernoulli observations) and clustered spatial locations. By 2016 all areas were estimated to have reached a low risk, although there is uncertainty in these estimates compared to P. vivax, due to the smaller sample size.

## Short–term predictions and temporal patterns in time series of *Plasmodium vivax* cases

Using an additive regression method to make short-term predictions, we estimate a posterior mean $R_c$ of 0.005 (95% CI = 0–0.34) for *Plasmodium vivax* cases in the year leading up to 2020 (Fig 4A). We observe an overall declining trend, with the fitted trend for $R_c$ (which estimates the general trend, separate to the influence of seasonal and holiday effects) declining from 0.31 (95% CI = 0.31, 0.34) at the start of 2011 to 0.004 (95% CI = 0.002–0.006) by the end of 2019

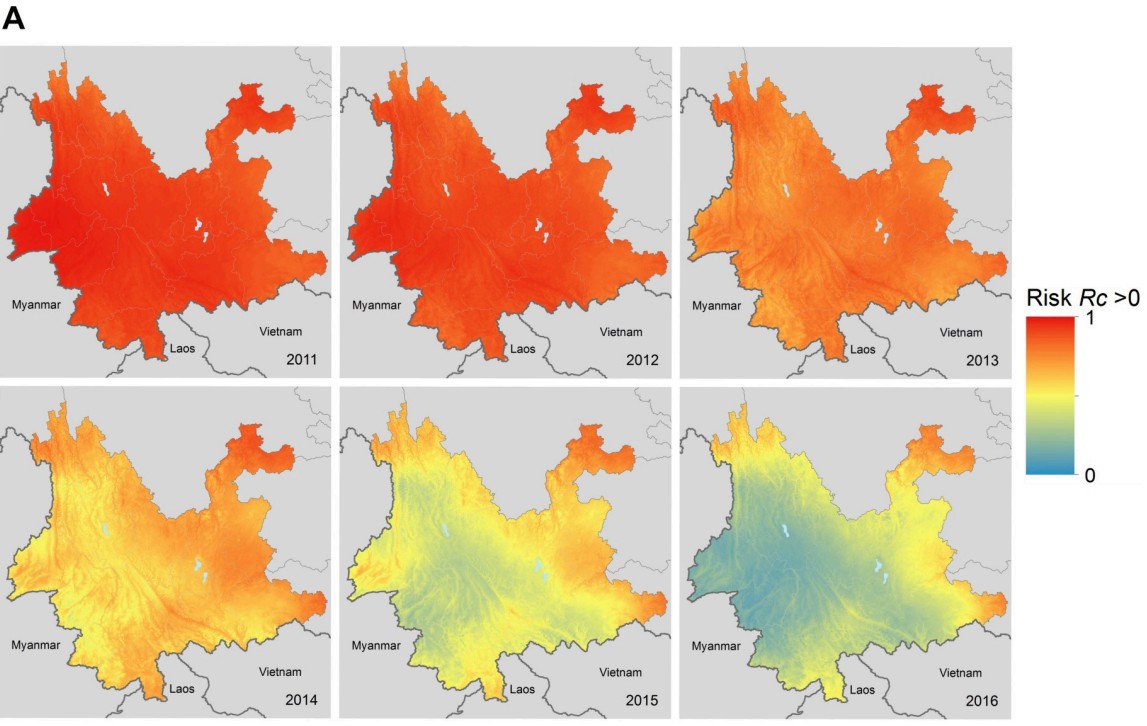

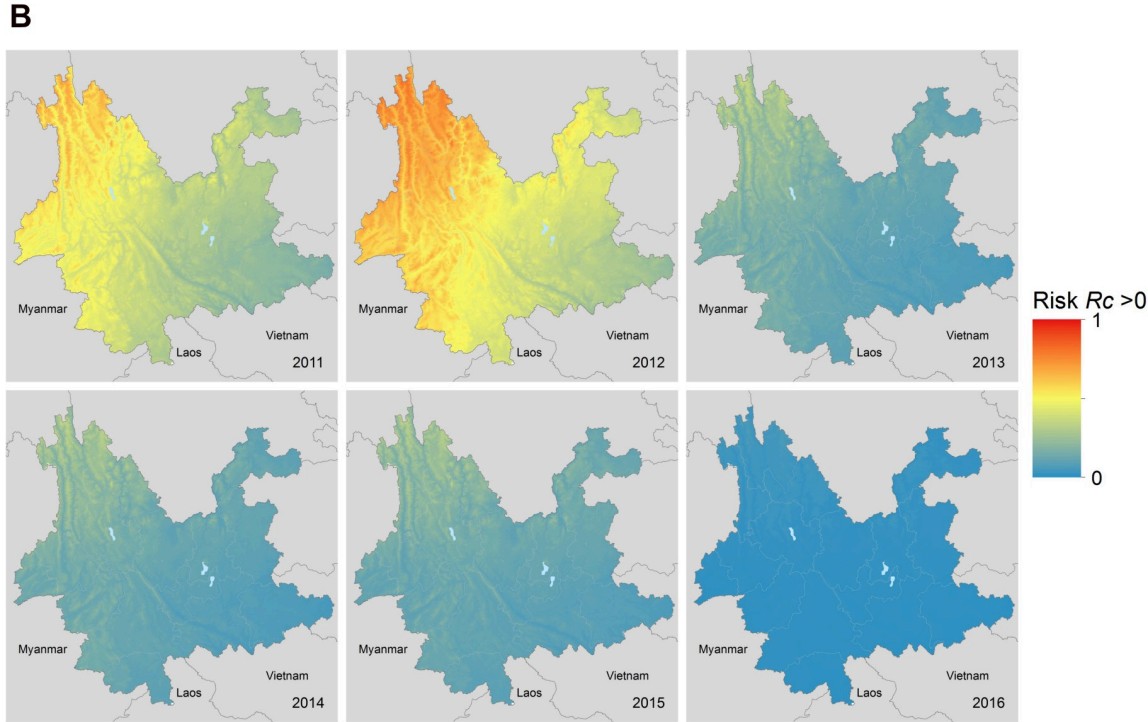

**Fig 3.** Map of risk of $R_c > 0$ and uncertainty in this estimate (standard deviation) from application of a Gaussian Process geostatistical model with a logit link function to times and locations of observed cases for A) P. vivax and B) P. falciparum malaria across Yunnan province in each year 2011–2016. This represents the risk of a case having an $R_c > 0$ if observed, stratified by year.

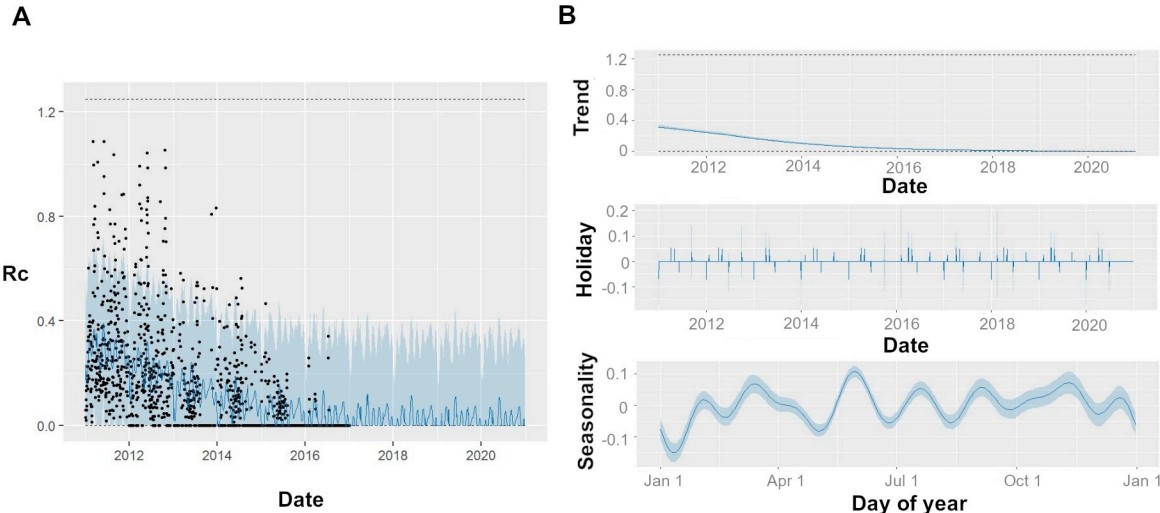

**Fig 4.** A) Black points show estimated individual $R_c$ values, blue line represents prophet model predictions for mean $R_c$ on that day, shaded blue area shows 95% credible interval of prediction. B) Decomposed time series model, showing the general trend, fitted holiday effect and seasonal effect. For seasonal and holiday effects the y axis shows the percentage increase or decrease in $R_c$ predicted which is attributable to a seasonal or holiday effect.

(Fig 4B). We estimate a small effect of holiday periods to differences in $R_c$ observed, with Chinese New Year and National Day associated with small increase risk in $R_c$ of 16% (95% CI = -112%, 152%) and 39% (95% CI = -43%, 118%) (Fig 4B) which in this very low transmission context could increase the probability of small outbreaks of local transmission in areas in which high rates of importation occur, although very wide credible intervals were associated with these estimates. We did not identify a clear seasonal trend, however two peaks were identified, with up to 20% (95% CI = 14%, 26%) increases and 28% decreases (95% CI -35%, -22%) in risk of $R_c$ associated with April/October and the beginning of January respectively (Fig 4B).

## Discussion

Quantifying reproduction numbers and their spatio-temporal variation can provide useful information to inform strategies to achieve and maintain elimination in contexts where traditional measures of transmission intensity are not appropriate. We used individual level surveillance data to infer reproduction numbers by estimating the likelihood of cases being linked by transmission and applied this to a dataset of all confirmed and probable cases of *P. vivax* and *P. falciparum* occurring in Yunnan province between 2011 and 2016, which is a focus of concern for re-emergence. Our results suggest that transmission in this province decreased rapidly between 2011 and 2016 as shown by a declining risk of $R_c$ exceeding zero across the province. This decline is relatively robust to assumptions about the serial interval distribution. Extrapolating this trend produces an estimated mean $R_c$ of 0.005 by 2020.

Given the consistently very low $R_c$ values estimated by 2014 onwards, and the future projections based on observed reproduction numbers over time, our results suggest that re-emergence or outbreaks of sustained transmissions are unlikely, provided interventions are continued. However, as all data analysed was collected whilst the NMEP was in place, we cannot draw conclusions about the impact of scaling back interventions or consider other counterfactuals. There is also some uncertainty in our estimates of current and future $R_c$, although the 95% credible intervals of these estimates remain below 1. It is important to note that even with low $R_c$ values it is still possible for locally-acquired cases to occur following importation,

however the probability of sustained chains of transmission decreases as $R_c$ decreases. There also is more uncertainty in our estimates of risk in areas that have not observed many cases. It is difficult to determine whether an absence of cases is due to a lack of detection, a lack of importation events occurring or a low underlying receptivity to transmission. However, it is worth noting that the greatest uncertainty in our spatiotemporal risk estimates of $R_c>0$ tends to be in areas of high elevation (elevation > 3000m), where there is unlikely to be transmission. Given the large numbers of imported cases, it is important to highlight these uncertainties and ensure control measures are maintained. Nonetheless, our findings are promising for China to meet their 2020 elimination goal. Our results highlight the success the country has had in malaria control and highlights the difficulty of elimination certification in contexts where both distant and local cross border importation is common.

The results of the additive regression model is suggestive of two moderate seasonal increases in $R_c$, one occurring in March/April, and one in October. This is a notably different pattern to a clear peak in the *incidence* of cases occurring in May (S4 Fig). This is interesting to note, as the March/April increase in $R_c$ would be expected to occur prior to an increase in incidence, however the second apparent peak in $R_c$ is not reflected by an increase in incidence in the months following. This pattern could be an artefact of human movement, with both periods associated with seasonal movement and holiday periods–the *Chunyun* period occurs in China for Chinese New Year and the holiday week of the National Day in October and is associated with intranational travel to visit family. During this time, there is often movement from cities to rural areas, and so in these contexts there may be more opportunities for infection to occur as more people are exposed to bites from suitable vectors. This is supplemented by our finding that these specific holidays are associated with small to moderate increases in $R_c$, however it is worth noting the very wide credible intervals and the great deal of uncertainty associated with these estimates, and therefore caution is required when interpreting this finding.

There are several limitations to our study. Firstly, there is a limitation in the classification of local and imported cases used in this study. For instance, the definition of importation used in case classification is defined by travel to any malaria-endemic areas outside China in the month prior to illness onset. This definition might include people who travelled abroad within the week prior to illness onset, but biologically their infection could not have been obtained during that time given the incubation period. However, in the absence of alternative information, travel history may provide a better indication of the likely importation status of a case than attempting to infer importation without this information, however there could be scope in future work to allow for incorrect travel history. As certification of elimination is now tolerant of introduced (first generation imported-to-local transmission) but not indigenous (second generation local-to-local transmission) cases, being able to differentiate between the two, and understanding how much transmission is indigenous versus imported or introduced is an important area of focus for future work.

It is important to consider unobserved cases and their potential contribution to transmission dynamics. We do account for unobserved cases via epsilon edges; however, this method is still more suited to scenarios where the majority of cases are observed. In contexts with a high level of asymptomatic infection contributing to transmission or with poor case detection and/or reporting, these approaches would not be suitable. However, in the context considered here there is good evidence of strong case reporting, investigation and response. The Ministry of Health (MoH) in China has been measuring the timeliness of the recommended protocol and ability to meet the 1-3-7 targets for surveillance, case investigation and management. It was found that the one-day target for case reporting was almost always met because this is required by law. In the years following the introduction of the 1-3-7 policy, the proportion of cases investigated within three days increased from roughly 55% in 2011 to almost 100% by 2013.

However the programme took longer to achieve the seven day focal point investigation goals, with just over 50% of foci investigated and treated within seven days by the end of 2013 [7]. Nevertheless, by 2015, adherence to the 1-3-7 strategy improved and this figure increased to an estimated 96% [24]. Whilst some cases could still be missed, the thoroughness of the approach means numbers of missing cases are likely to be small.

Treatment seeking behaviours also are important to consider for cases with mild or no symptoms, both in terms of not seeking treatment, or finding alterative treatment though self-medicating or informal providers. According to China CDC, Yunnan has achieved full coverage of basic healthcare at village/community level, and local residents and migrants can easily and timely access to health services. Following the National Malaria Elimination Action Plan, hospitals/CDCs provide free antimalarials to cure both local and imported malaria cases found in Chinese residents, mobile populations or foreigners, and pharmacies and non-medical institutions cannot sell antimalarials. Additionally, since China launched the Malaria Elimination Action Plan in 2010, Yunnan Province has special fund to support enhanced malaria surveillance with a high sensitivity, as it was identified as a key province with endemic malaria. The healthcare units at township level are required to screen malaria in febrile patients, so that the possibility of missed diagnosis or underreports of malaria in the patients with fever is likely to be low. Due to the low levels of malaria in China, immunity is likely to be low and therefore asyptomatic infection is less likely. For those who do have no fever or mild symptoms, we assume they have limited capacity to transmit malaria as their gametocyte density is likely to be low. Given the low levels of asymptomatic parasitaemia one would expect in this context, it is unlikely that these infections are contributing much to ongoing infection and therefore affecting our estimates, especially as we do estimate and allow for unobserved sources of infection in our approach.

A further consideration is the potential for relapse within *P. vivax* cases, and whether this may bias our estimates. According to China's National Malaria Elimination Technology Program (2011), the epidemiological history of each case has been investigated to check the source of the infection and the history of previous infection and relapse malaria. The individual-based data recorded electronically allow us to compare and find out the relapse malaria, but no relapse cases were found during the study. In addition, all malaria cases received free antimalarial treatment, and each case of P. vivax malaria was treated for radical cure. Nonetheless, there is still a small chance of some relapse cases. In our approach we jointly estimate unobserved sources of infection but are agnostic as to the specific cause of the unobserved source, relapses are considered as one of the potential unobserved sources of infection. Although large amounts of relapse are unlikely for the reasons outlined above, if there were very large amounts of relapse, the estimated reproduction numbers could be over-estimated. However, given we find such low reproduction numbers, even if this unlikely situation were the case, this does not impact our key findings and in fact would be stronger evidence of China achieving large reductions in malaria transmission between 2011 and 2016.

There is a limitation in the type of data available for inference. Although not available for this study, there are several data sources that increasingly are being collected and could enhance similar analyses in the future in eliminating and pre-eliminating contexts. Firstly, methods to make use of contact tracing data have been developed for emerging outbreaks [25] but have not to our knowledge been applied to endemic disease in the elimination. Although contact tracing for indirectly transmitted diseases is more difficult, identifying if the likely source of infection is a breeding site near the home or a place of work is carried out through active case detection schemes, but often the resulting data are not made available alongside line list data. This information could be used to weight certain connections. Genetic data are also increasingly available, and provide useful information about movement of parasites

[26,27], the likelihood of two cases being linked by transmission, and can provide useful information to help distinguish imported from local cases and chains of transmission resulting from importation from on-going local transmission [21]. Such data were not available in this context; however, a similar methodological framework or approach could incorporate information such as genetic distance. Historical data on incidence at fine scale (e.g. village level) could also be used to inform likelihood of asymptomatic infection.

We introduced a new framework for analysing individual level surveillance data and found that in Yunnan province, $R_c$ has seen a notable downward trend since 2011 and is expected to remain low with maintained interventions into 2020. This decline coincides with 1-3-7 strategy in improved adherence to guidelines. We predict a mean $R_c$ of 0.005 up to 2020, however even with such low $R_c$ values estimated, there may still be a need to continue to invest in detecting and rapidly responding to imported cases in order to achieve three consecutive years of zero cases and prevent resurgence. Nevertheless, China's elimination strategy and investment in surveillance provides a useful roadmap for other countries planning for malaria elimination by illustrating how coordinated and timely surveillance and response can be implemented, as well as sustained investment in surveillance, and region-focused international collaboration.

## Methods

### Ethics statement

It was determined by the National Health and Family Planning Commission, China, that the collection of malaria case reports was part of continuing public health surveillance of a notifiable infectious disease. The ethical clearance of collecting and using second-hand malaria data from the surveillance was also granted by the institutional review board of the University of Southampton, UK (No. 18152). All data were supplied and analysed in an anonymous format, without access to personal identifying information.

### Data

Anonymised case data for all confirmed (N = 4078) and probable (N = 285) malaria cases reported between 2011 and 2016 in Yunnan Province (N = 4390) were obtained from the Chinese Centre for Disease Control (CCDC). For each case, our data included date of symptom onset, GPS coordinates of symptom onset address, health facility address, travel history, and in some cases, the GPS coordinates of presumed location of infection.

Of these cases, the majority were *P. vivax* (N = 3469, of which 2858 were classified as imported). Of all recorded *P. falciparum* cases (N = 791), 91% (N = 720) were imported. Small numbers of *P. malariae* (N = 8) and *P. ovale* (N = 1) were excluded from our analysis. Cases defined as "untyped" (N = 67) were also excluded. A small number (N = 27) of cases classified as mixed infection were included in the separate analyses of each species. A full breakdown of the cases and species composition across China and in Yunnan province between 2011 and 2016 is included in S1 Table.

### Surveillance system in China

China has a sophisticated malaria surveillance system, described in detail elsewhere [7,15,16,24,28] and in S1 Text. Surveillance is carried out in both a passive and reactive manner, organised and administered at the national, provincial and county level. The centralised China Information System for Disease Control and Prevention (CISDCP) receives daily updates on case reports from health facilities.

Passive detection occurs according to a protocol at the local level, such that cases are tested by microscopy or Rapid Diagnostic Test (RDT) and reported to the central information system within 24 hours. Case investigation is then pursued, where cases are confirmed via double readings of microscopy slides and in some cases polymerase chain reaction (PCR) confirmation at provincial laboratories. At this point it is also determined whether the case is locally acquired or imported by taking patient travel history–if a patient has travelled to a malaria endemic country within a month of symptom onset the case is then classified as imported [7]. Case investigation should be completed within three days.

Foci investigation occurs once a case is detected to determine whether the foci is inactive, active or a pseudo-focus based upon the absence or presence of suitable vectors (inactive), and presence or absence of malaria in the resident area of the case if imported (pseudo-focus). Reactive Case Detection (RACD) of case contacts and populations with demographic similarities (for example individuals working in the same industry and vicinity as the case) is carried out. In active foci more intensive RACD screening of a larger pool of neighbours and contacts is carried out using Rapid Diagnostic Tests (RDTs) for immediate detection, followed by PCR testing of blood spots to detect low-density infections. IRS (Indoor Residual Spraying) is also carried out [6,7,24].

## Defining the serial interval distribution

The serial interval is defined as the time between a given case showing symptoms and the subsequent cases they infect showing symptoms [29]. For a given individual $j$ at time $t_j$, showing symptoms before individual $i$ at time $t_i$, the serial interval distribution specifies the normalised likelihood or probability density of case $j$ infecting case $i$ based on the time between symptom onsets, $t_i - t_j$. The serial interval summarises several distributions including the distribution of a) the times between symptom onset and infectiousness onset, b) the time for humans to transmit malaria parasites to mosquito vectors, c) the period of mosquito infectiousness, and d) the human incubation period.

Taking a similar approach to our previously developed work [20] using a greedy algorithm to estimate the reproduction numbers of malaria cases in El Salvador, we defined a prior distribution of possible serial interval distributions for malaria which were then used in the algorithm. The serial interval distribution of treated, symptomatic *P. falciparum* malaria, previously characterised using empirical and model based evidence [30,31] was adapted to inform the prior distribution for the relationship between time and likelihood of transmission between cases in China. We analysed *P. vivax* cases and *P. falciparum* cases separately. The prior distribution was defined to be flexible enough to reflect both the biology of *P. vivax* and *P. falciparum* as well as the dominant vector species in Yunnan (recent surveys in Yunnan province have found *Anopheles sinensis* to be the dominant vector species in mid-elevation areas and rice paddies and *Anopheles minimus* s.l. the dominant species in low elevation areas [13,32] and to allow for possible variation in transmission dynamics, for example due to untyped infections or delays in seeking treatment. In addition, there is a possibility of a small number of asymptomatic or undetected and therefore untreated infections contributing to ongoing transmission, which will typically have a longer serial interval. We use a shifted Rayleigh distribution (a special case of a Weibull distribution), to describe the serial interval of both species, which can be varied by changing two parameters: $\alpha$ and $\gamma$. The parameter $\alpha$ governs the overall shape of the distribution, and $\gamma$ is the shifting parameter accounting for the incubation period between receiving an infectious bite and the onset of symptoms (Fig 1A). The $\gamma$ shifting parameter was fixed at 15 days to account for the extrinsic incubation period within the mosquito and the minimum time between infection and suitable numbers of gametocytes in the blood to lead to symptom onset [33]. The prior for the $\alpha$ parameter determining

## Serial Interval Distribution

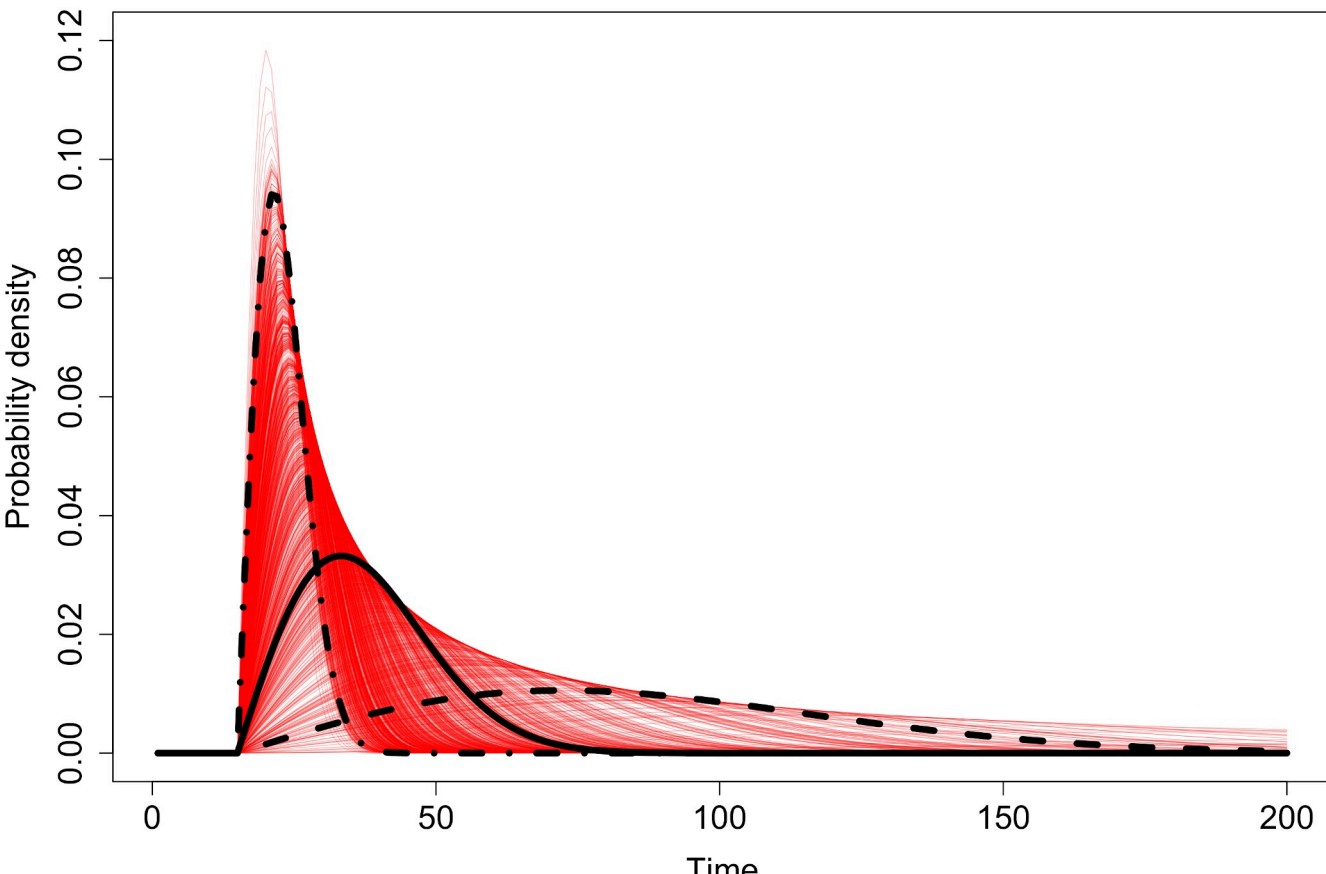

**Fig 5. Red lines show 300 draws from the prior distribution used in the analysis for the Serial Interval distribution.** The black solid line represents the expected function and the dashed lines represents the 2.5 (dashed) and 97.5 (dot-dashed) quantile values of the prior distribution for the shaping parameter, α.

the shape of the distribution was given a Normal distribution with mean 0.003 and standard deviation 0.1 (illustrated in Fig 5), giving an expected time between symptom onset of one case and symptom onset of the case it infects of 36 days, with the parameter value in the 2.5 percentile of prior having an expected serial interval of 21 days and the equivalent parameter from the 97.5 percentile having an expected serial interval of 60 days. By comparison the expected values for treated *P. falciparum* from existing literature range between 33 and 49.1 days (95% CI = 33–69) [30,31]. Depending on how much uncertainty there is in the serial interval (SI) of malaria, the prior for α, the shaping parameter for the SI of malaria, may be varied. We explored the effects of different priors on the likelihood and posterior estimates. We used the same mean value for α (0.003) but set the α prior to standard deviation between 1 and 0.01. The results of considering different priors for α, the parameter shaping SI distribution on estimated $R_c$ values over time is shown in S2 Fig.

### Determining the transmission likelihood

We assume cases were classified correctly from case investigation as imported or locally-acquired based on recent travel history. Following this assumption, locally-acquired cases

could have both infected others and been infected themselves. However imported cases could only infect others, as we assume their infection was acquired outside of the country. Given the evidence [7,15,24] of strong adherence to the 1-3-7 policy for reporting and response to case detection, and no evidence of relapse within the dataset (as each patient is given a unique identifier), we assume that an individual can only be infected once by a case that has shown symptoms earlier in time.

### Transmission model specifics

To estimate the underlying pathways of transmission and likelihood of cases being linked by infection, we adapt and extend the NetRate algorithm [22]. Our adapted model introduces the ability to model serial interval functions, account for imported vs local infections and provides provision for missing or unobserved sources of infection (called epsilon edges [20,34]). We also extended the NetRate algorithm from a frequentist to a Bayesian framework to incorporate prior knowledge about the serial interval of malaria.

Consider a set of $n$ infections/nodes $I \in (I_1, \ldots, I_n)$ with associated times $t = \{t_1, \ldots t_n\} \in \mathbb{R}^+$ and binary yes/no importation status $\pi = \{\pi_1, \ldots, \pi_n\} \in \{1, 0\}$. The serial interval distribution of malaria, defining the probability individual $I_j$ infected individual $I_i$ at times $t_i > t_j$ is defined through a shifted Rayleigh distribution $f(t_i | t_j; \alpha, \gamma) = \alpha(t_i - t_j - \gamma)e^{-\alpha(t_i - t_j - \gamma)}$ for shaping parameters $\alpha$ and $\gamma$ [20]. For our analysis we fix $\gamma = 15$ days. If we assume that infections are conditionally independent given the parents of infected nodes then the likelihood of a given transmission chain can be defined as

$$f(\boldsymbol{t}; \boldsymbol{\alpha}) = \prod_{t_i \in t} f(t_i | t_1, \ldots, t_n \setminus t_i; \boldsymbol{\alpha}) \tag{1}$$

Where $\boldsymbol{\alpha}$ is a parameter matrix. Computing the likelihood of a given transmission chain thus involves computing the conditional likelihood of the infection time of each infection ($t_i$) given all other infections ($t_1, \ldots, t_n \setminus t_i$). If we make the assumption that a node gets infected once the first parent infects it [35] and define a survival function

$$S(t_i | t_j; \alpha_{j,i}) = 1 - \int_0^{t_i - t_j} f(t_i | t_j; \alpha_{j,i})dt \tag{2}$$

as the probability that infection $I_i$ is <u>not</u> infected by infection $I_j$ by time $t_i$ then we can simplify our transmission likelihood as

$$f(\boldsymbol{t}; \boldsymbol{\alpha}) = \prod_{t_i \in t} \sum_{I_j : t_j < t_i} f(t_i | t_j; \alpha_{j,i}) \prod_{I_k : t_k < t_i, I_k \neq I_j} S(t_i | t_k; \alpha_{k,i}) \tag{3}$$

In this conditional likelihood the first term computes the probability the $I_j$ infected $I_i$ and the second term computes the probability that $I_i$ was not infected by any *other* previous infections excluding $I_j$. This likelihood therefore accounts for competing infectors and finds the infector most likely to have infected $I_i$. To remove the $k \neq j$ condition makes the product independent of $j$ and results in the likelihood

$$f(\boldsymbol{t}; \boldsymbol{\alpha}) = \prod_{t_i \in t} \prod_{I_k : t_k < t_i} S(t_i | t_k; \alpha_{k,i}) \sum_{I_j : t_j < t_i} \frac{f(t_i | t_j; \alpha_{j,i})}{S(t_i | t_j; \alpha_{j,i})} \tag{4}$$

In Eq 4, $f(\cdot) / S(\cdot) = H$ is the hazard function or rate and represents the instantaneous infection rate between individuals $I_i$ and $I_j$.

Assuming all cases reaching health workers or health facilities are recorded, missing cases may be generated by two processes. Symptomatic cases may be missed by not seeking care or not being found through active case detection, or cases may be asymptomatic and therefore unlikely to seek care or be detected. The latter may have densities of parasites in their blood

which are too low to be detectable by microscopy if active case detection occurs. These processes apply to both imported cases or locally acquired cases. We assume the pool of asymptomatic cases in the country is low and has a small contribution to ongoing transmission. To account for unobserved infectors within our framework we include a time-independent edge that can infect any individual. The survival and hazard functions for this edge are defined as $S_0(\epsilon_i) = e^{-\epsilon_i}$ and $H_0 = \epsilon_i$. As we will see below, as a consequence of our optimisation problem these edges are encouraged to be sparse and only invoked if no other infectors can continue the transmission chain.

In addition to unobserved edges, we assume that observed imported infectors can infect other cases but cannot be infected themselves. The final likelihood incorporating these two modifications becomes

$$f(\boldsymbol{t}; \boldsymbol{\alpha}, \epsilon) = \prod_{t_i \in \boldsymbol{t}} S_0(\epsilon_i) \prod_{I_k : t_k < t_i} S(t_i | t_k; \alpha_{k,i})(H_0(\epsilon_i) + \sum_{I_j : t_j < t_i} H(t_i | t_k; \alpha_{k,i})) \qquad (5)$$

In order to find the optimal parameters for $\boldsymbol{\alpha}, \epsilon$ we minimize the following log likelihood subject to positive constraints on the parameters:

$$minimize_{\boldsymbol{\alpha}, \epsilon} - \log f(\boldsymbol{t}; \boldsymbol{\alpha}, \epsilon) \quad subject \ to \ \boldsymbol{\alpha}, \epsilon > 0 \ \forall i, j \qquad (6)$$

This optimisation problem is convex and guarantees a consistent maximum likelihood estimate [22]

To prevent biologically implausible serial interval distributions we impose a truncated normal prior probability distribution on $\boldsymbol{\alpha}$ ~Normal(0.003,0.1) [0,0.01]. When optimising our likelihood we include this prior probability and therefore evaluate the Bayesian Maximum-a-Posteriori estimate.

## Estimating $R_c$

We can establish individual reproduction numbers for each case by creating a matrix where each column represents a potential infector and the rows represent a potential infectee, describing which infector edges are connected to infectees and the normalised likelihood of the cases being connected by a transmission event. Intuitively then, by taking the row sums we get the (fractional) number of secondary infections and therefore a point estimate of the time varying reproduction number $R_c(t_j)$ This reflects for an individual, an expectation of how many people they are likely to have subsequently infected. When multiple individuals have been infected at a given time and/or place, we can take the mean individual $R_c$ and uncertainty in this value as an indicator of reproduction numbers for a given time and/or location.

In contrast to traditional methods based on Wallinga and Teunis [36] using our method in this way encapsulates not only the pairwise likelihood of transmission between two cases, but conditions this likelihood on the impact of competing edges in the inferred network (the survival of an edge). Our estimates of $R_c$ therefore consider the overall transmission tree in optimisation and allow for missing cases within the tree.

## Simulating data

In order to test the ability of the approach to recover the mean and distribution of $R_c$ estimates, and evaluate the impact of missing cases, we carried out two simulations, both simulating a susceptible infected model over a small world network and generating simulated datasets using a stochastic SEIR model with a negative binomial distribution of $R_c$ values of mean 0.5, and two different values for the overdispersion parameter, $K(K = 0.1$ and $K = 1)$, which

represents the amount of variability in individual reproduction numbers. The simulation regime and results are described fully in S2 Text.

## Estimating timelines towards elimination and risks of resurgence

It is important for national malaria control programmes to have information about likely time-lines to elimination, chances of resurgence and uncertainty in these estimates. Using the distribution of $\mathcal{R}_c$ values and their seasonal and general trends, we analysed time series using the *Prophet* tool and R package [37] to explore general and seasonal trends as well as the impact of holidays on results.

This approach applies an additive regression model

$$y(t) = g(t) + s(t) + h(t) + \epsilon_t \tag{7}$$

which is composed of trend, seasonal and holiday functions, where $y(t)$ is the observations at time $t$, $g(t)$ is the general trend, modelled by a logistic growth model, $s(t)$ is the seasonal effect, modelled by Fourier series, $h(t)$ is the effect of specific holiday dates and $\epsilon_t$ is the error term.

We explored the overall trend as well as seasonal trends, exploring the predicted $R_c$ between 2011 and the beginning of 2020. We also explored the impact of the national holiday periods, some of which involve large scale movement, such as the *Chunyun* period around the spring festival. We cross-validated predictions and calculated root mean squared error (RMSE) and mean absolute error (MAE) (S6 Fig).

## Mapping $R_c$

Transmission risk map estimates were constructed by separating individual reproduction numbers into those above and below $R_c = 1$ The latitude and longitude of the reproduction numbers were included in a binomial Gaussian random field model implemented in rINLA [38], described in the geostatistical model specifics section, in which demographic and environmental covariates (Table 1) were used to estimate the likelihood of a case having $R_c>0$ in the area each year from 2011 to 2016.

We chose covariates which have previously been found to be strongly mechanistically linked to malaria transmission due to their impact on vector habitat and exposure risk (Tables 1, 2 and 3). These covariates have also previously been found to predict malaria prevalence [39].

This is a measure of malaria "receptivity" or underlying transmission potential rather than overall malaria risk, as importation likelihood is not quantified in this analysis. An auto-regressive model of order 1 was also fitted to the binomial data and the distance range of spatial correlation in $R_c>0$ were also investigated to identify the amount of focality in the distribution of

**Table 1. Environmental and demographic covariates used in geostatistical analysis.** First column lists the class of variable, the second column lists the variables used, the third column lists sources for the data, the fourth column lists the type of data (static, synoptic or dynamic), and the final column lists the spatial scale of the data used to generate the variables.

| Variable Class | Variable(s) Source Type | | | Spatial scale |
|---|---|---|---|---|
| Temperature | land surface temperature (day, night and diurnal-flux) | MODIS product [40] | dynamic monthly | 2.5 arc-minute (~5 km x 5 km) |
| Precipitation | mean annual precipitation | WorldClim [41] | synoptic | 2.5 arc-minute (~5 km x 5 km) |
| Elevation | digital elevation model | SRTM [42] | static | 2.5 arc-minute (~5 km x 5 km) |
| Infrastructural development | accessibility to urban centres and night-time lights | modelled product and VIIRS [43] | static | 2.5 arc-minute (~5 km x 5 km) |

**Table 2. Table summarizing posterior parameter estimates for covariates in geostatistical model for A) P. vivax and B) P. falciparum.** Columns show posterior mean, standard deviation and 2.5, 97.5 and 50% quantiles. Rows represent covariates used in model and intercept.

| Covariate | Mean | SD | 0.025 Quantile | 0.5 Quantile | 0.975 Quantile | Mode |
|---|---|---|---|---|---|---|
| Elevation | -0.00065 | 0.000369 | -0.00137 | -0.00065 | 7.82E-05 | -0.00065 |
| Day temperature (monthly) | 0.040258 | 19.13436 | -37.5269 | 0.03972 | 37.57611 | 0.040258 |
| Night temperature (monthly) | -0.11265 | 19.13447 | -37.6801 | -0.11319 | 37.42342 | -0.11265 |
| Difference between day and night-time temperature (monthly) | -0.07346 | 19.13443 | -37.6408 | -0.074 | 37.46253 | -0.07346 |
| Precipitation | -0.00041 | 0.000248 | -0.00089 | -0.00041 | 7.96E-05 | -0.00041 |
| Urban | -0.06908 | 0.301495 | -0.66102 | -0.06909 | 0.522361 | -0.06908 |
| Intercept | 4.065973 | 1.985619 | 0.167532 | 4.065917 | 7.961159 | 4.065973 |

**Table 3. Posterior covariate parameter estimates for _P. falciparum_ $R_c$ risk map.**

| Covariate | Mean | SD | 0.025 Quantile | 0.5 Quantile | 0.975 Quantile | Mode |
|---|---|---|---|---|---|---|
| Elevation | 0.000112 | 0.000502 | -0.00087 | 0.000112 | 0.001097 | 0.000112 |
| Day temperature (monthly) | -0.01005 | 19.12776 | -37.5643 | -0.01059 | 37.51285 | -0.01005 |
| Night temperature (monthly) | -0.03118 | 19.12771 | -37.5853 | -0.03172 | 37.49163 | -0.03118 |
| Difference between day and night-time temperature (monthly) | -0.00245 | 19.12769 | -37.5566 | -0.00299 | 37.52031 | -0.00245 |
| Precipitation | 0.00015 | 0.00029 | -0.00042 | 0.00015 | 0.000718 | 0.00015 |
| Urban | 0.361755 | 0.452532 | -0.52672 | 0.361743 | 1.249487 | 0.361755 |
| Intercept | -1.86989 | 3.045358 | -7.84895 | -1.86998 | 4.104185 | -1.86989 |

non-zero $R_c$ estimates. Area under the curve (AUC) scores from leave-one-out cross validation were used to assess model fit (S7 Fig).

## Geostatistical model specifics

The underlying spatial statistical model was fitted to binomial data of $R_c > 0 = 1$; $R_c = 0$, using the logit link function:

$$R_{>0,i}^+ \sim Binomial(p_i, N_i)$$

$$log(p_i/(1 - p_i)) \sim GP(\mu, Q)$$

$$\mu = \alpha + X_i\beta$$

$$Q = K_{space}^{-1}$$

$$K_{space}^{-1} = solve(k^2 - \Delta)^{\frac{z}{2}}(\tau x(s)) = W(s)$$

where $R_{>0,i}$ are the number binary data points for $R_c > 0 = 1$; $R_c = 0$, $p_i$ is the estimated $R_{>0}$, expressed as a logit transformed probability and modelled as a Gaussian process with $\mu$ and precision Q. The GP mean $\mu$ is a linear function of a global intercept $\alpha$ and a vector of $\beta$ coefficients from space-time indexed covariate values $X_i$. Q is a sparse precision matrix constructed as the sparse finite element solution to the stochastic partial differential equation $(k^2 - \Delta)^{\frac{z}{2}}(\tau x(s)) = W(s)$, where $\Delta$ is the Laplacian, $k$ is the spatial scale/range parameter, $\tau$ controls the variance, $\alpha$ is the spatial smoothness parameter (fixed at $\alpha = 2$), and $W(s)$ is the spatial white noise process. To account for the curvature of the earth the distance metric $s$ is defined on a spherical manifold in Cartesian $\mathbb{R}^3$.

## Supporting information

**S1 Fig. Histogram of $R_c$ estimates for A)** *P. vivax* **in Yunnan and B)** *P. falciparum* **in Yunnan.** Dotted lines show median, solid lines show mean.
(PNG)

**S2 Fig. Maximum a posteriori estimates of $R_c$ over time for A)** *P. falciparum* **and B)** *P. vivax* **in Yunnan province.** Colours represent different standard deviations of prior for α, the shaping parameter for the serial interval distribution.
(PNG)

**S3 Fig. $R_c$ by month compared to cases.** Stratified by year-month for *P. vivax*.
(PNG)

**S4 Fig.** Histogram of epsilon edge distribution for A) *P. vivax*, B) *P. falciparum*.
(PNG)

**S5 Fig.** Map of standard deviation in estimates of P($R_c$ >0) for A) *Plasmodium falciparum* and B) *Plasmodium vivax*.
(PNG)

**S6 Fig. Cross validation of timeseries analysis.** Plots show A) Root Mean Squared Error (RMSE) B) Mean Absolute Error (MAE). The training set used was the first 730 days of data and the horizon used was 365 days.
(PNG)

**S7 Fig.** ROC curve plot for map of P($R_c$ risk >0) for A) *P. vivax* B) *P. falciparum*.
(PNG)

**S1 Text. Surveillance system in China.**
(DOCX)

**S2 Text. Tests of algorithm on simulated data.**
(DOCX)

**S1 Table. Table showing breakdown of case data.** A) Cases by diagnosis type (probable and confirmed) and species across China. B) Cases by imported/local status and species across China. C) Cases by diagnosis type (probable and confirmed) and species across Yunnan Province. D) Cases by imported/local status and species across Yunnan province2
(DOCX)

**S1 Data. Source code to generate figures, data listing estimated Rc values and results of additive regression analysis from Fig 4.**
(ZIP)

## Author Contributions

**Conceptualization:** Isobel Routledge, Shengjie Lai, Azra C. Ghani, Andrew J. Tatem, Samir Bhatt.

**Data curation:** Isobel Routledge, Shengjie Lai, Zhongjie Li.

**Formal analysis:** Isobel Routledge.

**Funding acquisition:** Isobel Routledge.

**Investigation:** Isobel Routledge, Shengjie Lai, Samir Bhatt.

**Methodology:** Isobel Routledge, Manuel Gomez-Rodriguez, Kyle B. Gustafson, Swapnil Mishra, Joshua L. Proctor, Samir Bhatt.

**Project administration:** Isobel Routledge.

**Resources:** Shengjie Lai.

**Supervision:** Azra C. Ghani, Manuel Gomez-Rodriguez, Andrew J. Tatem, Zhongjie Li, Samir Bhatt.

**Visualization:** Isobel Routledge, Katherine E. Battle.

**Writing – original draft:** Isobel Routledge.

**Writing – review & editing:** Isobel Routledge, Shengjie Lai, Katherine E. Battle, Azra C. Ghani, Manuel Gomez-Rodriguez, Kyle B. Gustafson, Swapnil Mishra, Juliette Unwin, Joshua L. Proctor, Andrew J. Tatem, Samir Bhatt.

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
