## [Decision Letter · Decision Letter 0]

24 Nov 2019

Dear Dr Routledge,

Thank you very much for submitting your manuscript, 'Tracking progress towards malaria elimination in China: individual-level estimates of transmission and its spatiotemporal variation using a diffusion network approach', to PLOS Computational Biology. As with all papers submitted to the journal, yours was fully evaluated by the PLOS Computational Biology editorial team, and in this case, by two independent peer reviewers. The reviewers found the work interesting and compelling but identified some aspects of the manuscript that should be improved.

We would therefore like to ask you to modify the manuscript according to the review recommendations before we can consider your manuscript for final acceptance. Your revisions should address the specific points made by each reviewer and we encourage you to respond to the particular issues, especially the point raised on relapses and how these would modify Rc. 

Please note while forming your response, if your article is accepted, you may have the opportunity to make the peer review history publicly available. The record will include editor decision letters (with reviews) and your responses to reviewer comments. If eligible, we will contact you to opt in or out.raised.

- Supporting Information uploaded as separate files, titled 'Dataset', 'Figure', 'Table', 'Text', 'Protocol', 'Audio', or 'Video'.

We hope to receive your revised manuscript within the next 30 days. If you anticipate any delay in its return, we ask that you let us know the expected resubmission date by email at ploscompbiol@plos.org.

Sincerely,

Mercedes Pascual

Associate Editor

PLOS Computational Biology

Virginia Pitzer

Deputy Editor

PLOS Computational Biology

[LINK]

Reviewer's Responses to Questions

**Comments to the Authors:**

Reviewer #1: General comments / Major issues

This is a well written paper on an important topic in malaria elimination research, namely using a diffusion network approach to estimate Rc for Plasmodium vivax and falciparum cases in Yunnan province between 2011-2016. The study methods and procedures are described adequately. There are a few areas that authors might consider for improving the paper.

Major compulsory comments to be addressed:

• The evidence for lack of a P. vivax relapse is stated to be from the surveillance dataset as each patient is given a unique identifier. I would argue this does not preclude there to be a fair number of relapsed P. vivax cases. How would this have influenced the estimation of Rc? I’m assuming it would have had limited impact on these estimates, but the authors should discuss this in the discussion.

• The authors argue there are a limited number of unobserved cases in the population because of strong reporting, investigation and response. What about treatment seeking in the community, where even with mild/non-severe symptoms, an individual either does not seek treatment, or finds alterative treatment though self-medicating or through informal providers? Are there any data on treatment seeking for fevers in Yunnan? I would be particularly concerned if lack of treatment seeking were also associated with certain populations that would be more likely to travel to malaria endemic settings, such as migrants or undocumented immigrants, who also might be at increased risk of infection and importation (and being an unobserved case). How might such unobserved cases have impact spatio-temporal trends in Rc? This would also be good to bring up in the discussion.

• In trying to untangle how peaks in Rc correspond to peaks in cases, are there any season trends in Rc and cases in neighboring malaria endemic countries/areas that might help explain these season trends in Yunnan?

Reviewer #2: This is an interesting and well-written study, detailing a methodology used to quantify spatio-temporal variation in reproduction numbers for malaria, using a dataset of of P. vivax and P. falciparum cases occurring in Yunnan province in China between 2011 and 2016. The methodological approach appears thorough and the results are on the whole well presented. I have the following minor comments and suggestions:

General

Line 59: provide more details about the malaria elimination plan. Did this include changes in prophylaxis or vector control? Has China done anything different compared to other countries to achieve success?

Line 132: discuss probable sources of uncertainty.

Line 136: “almost certainly due to the smaller sample size” —> remove “almost certainly” or suggest other reasons.

Methods

Line 305: should j and i be reversed, e.g. case i infecting case j?

Line 321: briefly describe what a Rayleigh distribution is

Line 334: define SI on first mention

Line 309: provide more details about ref 20.

Line 359: note different reference style.

Line 362: is the backslash symbol intended?

Geostatistical model specifics

How was the geospatial model spatially and temporal structured? Include space-time indicators/explanations where relevant. What are the assumptions about chosen environmental covariates and R_c? explain. Show parameter estimates for covariates.

Figures and Table

Figure 2: change legend title from imported to ‘source of infection’ (or something along those lines), change blue label to ‘locally acquired’ and red label to ‘imported’.

In the supplementary information, consider showing a map with a similar format to Figure 2 showing the case data that informed the model.

Figure 5: use different line type for 2.5 and 97.5 percentile values.

Table 1: include spatial and temporal resolution of data. Provide more details for data source, including references. Capitalise first letter.

Non-exhaustive list of typos/suggested edits:

Line 80: it can be unclear whether —> it is unclear if

Line 119: 0.5 <= epsilon <= 0.8

Line 129: missing fullstop after (Figure 2)

Line 151: and —> an

**Have all data underlying the figures and results presented in the manuscript been provided?**

Reviewer #1: Yes

Reviewer #2: Yes

PLOS authors have the option to publish the peer review history of their article (what does this mean?). If published, this will include your full peer review and any attached files.

Reviewer #1: Yes: Thom Eisele

Reviewer #2: No

---

## [Editor Report · Decision Letter 1]

3 Feb 2020

Dear Ms Routledge,

We are pleased to inform you that your manuscript 'Tracking progress towards malaria elimination in China: individual-level estimates of transmission and its spatiotemporal variation using a diffusion network approach' has been provisionally accepted for publication in PLOS Computational Biology.

Before your manuscript can be formally accepted you will need to complete some formatting changes, which you will receive in a follow up email. A member of our team will be in touch within two working days with a set of requests.

Best regards,

Mercedes Pascual

Associate Editor

PLOS Computational Biology

Virginia Pitzer

Deputy Editor

PLOS Computational Biology

---

## [Editor Report · Acceptance letter]

4 Mar 2020

PCOMPBIOL-D-19-01618R1 

Tracking progress towards malaria elimination in China: individual-level estimates of transmission and its spatiotemporal variation using a diffusion network approach

Dear Dr Routledge,

I am pleased to inform you that your manuscript has been formally accepted for publication in PLOS Computational Biology. Your manuscript is now with our production department and you will be notified of the publication date in due course.

With kind regards,

Laura Mallard
